# A Shorter Form of the Work Extrinsic and Intrinsic Motivation Scale: Construction and Factorial Validation

**DOI:** 10.3390/ijerph192113864

**Published:** 2022-10-25

**Authors:** Yasuhiro Kotera, Muhammad Aledeh, Annabel Rushforth, Nelly Otoo, Rory Colman, Elaina Taylor

**Affiliations:** 1School of Health Sciences, University of Nottingham, Nottingham NG7 2TU, UK; 2Klinik Donaustadt, Wiener Gesundheitsverbund, Langobardenstraße 122, AT-1220 Vienna, Austria; 3College of Health, Psychology and Social Care, University of Derby, Derby DE22 1GB, UK; 4Department of Human Resources and Administration, Khemas Care Partners, Carson, CA 90746, USA

**Keywords:** Work Extrinsic and Intrinsic Motivation Scale, work motivation, intrinsic motivation, extrinsic motivation, amotivation, short scale, scale construction, factorial validation

## Abstract

While workplace mental health has attracted attention in many countries, work motivation remains under-researched. Research identified that work motivation is associated with many organisational positive outcomes including workplace mental health. One well-recognised measure is the Work Extrinsic and Intrinsic Motivation Scale (WEIMS). Conceptualised on the Self-Determination Theory, this 18-item scale examines six types of work motivation: Intrinsic Motivation, Integrated Regulation, Identified Regulation, Introjected Regulation, External Regulation, and Amotivation. WEIMS can be too long for busy people at work. Accordingly, we constructed and validated a shorter form of WEIMS (SWEIMS), comprising 12 items that evaluate the same six work motivation types. Data collected from two professional samples were analysed to construct and validate the factorial structure: 155 construction workers (138 males and 17 females, Age 40.28 ± 11.05) and 103 hospitality workers (47 males and 56 females, Age 28.2 ± 8.6 years). Correlation analyses and confirmatory factor analyses were performed. Two items from each type were selected based on the strength of correlations with the target WEIMS subscale. SWEIMS demonstrated adequate internal consistency (α ≧ 0.65), and strong correlations with the original version of WEIMS (*r* = 0.73) in both samples. SWEIMS confirmatory factor analysis replicated the six-factor model of the original SWEIMS. SWEIMS can be a reliable, valid, and user-friendly alternative to WEIMS.

## 1. Introduction

### 1.1. Work Motivation as Key for Workplace Mental Health

Work motivation is in essence why individual employees engage in work, and the psychological factors that facilitate this [1]. Employees with high levels of work motivation perform better in their roles [2], and can significantly increase the productivity of the organisation [3]. Low work motivation has been associated with absenteeism [4,5] and poor goal achievement [6]. Work motivation is closely linked to employee satisfaction [7] and psychological wellbeing [8], and positively associated with autonomy and feelings of social relatedness [9]. Additionally, work motivation has a strong positive association with psychological empowerment [10]. A positive change in work motivation was significantly related to an improvement in employee exhaustion, and that a negative change in work motivation was related to both an increase in exhaustion and depression [11].

More than 10 million working days are lost as a result of employee stress, depression and anxiety per year, with an annual cost of GBP 10 billion to the UK economy [12]. Stress related illnesses affected 22% of EU workers [13]. A systematic review reported high emotional exhaustion (20–81%) in Arab world workers [14], and 1.4% of Korean workers experienced work-related depression [15]. Significant correlations were found between work motivation and mental health problems. Those with lower work motivation tended to experience work-related mental health problems compared to those with higher work motivation [16,17,18,19]. Additionally, issues with work motivation were a significant explanatory variable for depression, anxiety, and stress, accounting for 34–50% of the variance in these issues. Work motivation is important to work mental health [20]. 

### 1.2. Intrinsic and Extrinsic Motivation

Self-Determination Theory (SDT) is one of the most established motivation theories [21]. SDT differentiates people’s motivation in terms of either being autonomous or controlled [22]. The most autonomous motivation is regarded as intrinsic motivation, whereas the most controlled motivation is regarded as extrinsic motivation. Additionally, no motivation to work is understood as amotivation. 

Intrinsic motivation refers to people participating in activities in which the motivation for doing so lies in the behaviour or act itself [23]. Intrinsic motivation is that motivation when workers are motivated to do what they find interesting and enjoyable, and it is often associated with better mental well-being [24]. Intrinsic motivation is often expressed as passion for work. Intrinsically motivated workers feel that work activities themselves are already a reward for them. In contrast, extrinsic motivation is when employees work because it leads to some external rewards such as money and/or fame [25]. In general, intrinsic motivation is associated with positive organisational outcomes including good mental health, whereas extrinsic motivation is associated with negative ones such as poor mental health and shame towards mental health problems [26,27]. 

Extrinsic motivation can be further categorised into four subtypes: Integrated regulation, identified regulation, introjected regulation and external regulation (in the order of autonomy). Integrated regulation arises when an employee has fully integrated a motivation within themselves. They work because they believe work is part of their identity, that is, who they believe they are. Identified regulation relates to employees who acknowledge the value of the work activity. They work because they understand that matters to them. Introjected regulation is present when workers are motivated by self-image. They engage in work activities because they want other people to see them in a certain way. Lastly, an employee with external regulation works only because that brings them an external reward [28].

Amotivation refers to no motivation to work at all. Amotivation can occur for example when an employee does not believe that they can perform in ways required of them [28]. Amotivated employees often exhibit a low level of mental health and shame towards their own weaknesses [29].

### 1.3. Measuring Work Motivation

The Work Extrinsic and Intrinsic Motivation Scale (WEIMS) is one of the most established work motivation scales incorporating measurement of SDT motivation constructs [30]. Based upon the francophone 31-item Blais Inventory of Work Motivation scale [31], the 18-item WEIMS provides an anglophone measure that has been widely validated in various organisational settings [32], throughout diverse world regions [33,34] including recent cross-cultural studies [27,35]. Although some evidence has questioned the SDT continuum model underlying the WEIMS (e.g., the nature of each motivation is different thus is not linear) [36], its reliability and validity across wide-ranging organisational contexts positions it as a robust instrument to assess workplace motivation.

Alternative measures of work motivation informed by SDT either have poor psychometric qualities [37], fail to include the full range of SDT constructs [32], or contain more items than the WEIMS, such as the Multidimensional Work Motivation Scale [38]. This latter scale has addressed criticism of the WEIMS and other work motivation measures over the construct validity of certain items [38]; however, being less established, has less extensive evidence of ecological validity. 

Although the WEIMS was originally developed to provide a concise measure suited to efficient deployment in work environments [32], the pressures of modern organisational practices, including increased demand on cognitive resources [39], enhance the need to rationalise scales to their essential elements to reduce the time required to complete them [40]. Research in Internet mediated questionnaire design shows that shorter questionnaires reduce participant attrition [41,42], improving the representation of participants across diverse response profiles and consequent validity of findings [43]. By attending to factor structure, a parsimonious shortened version of the WEIMS, suitable for contemporary organizational settings, may appropriately be developed [44].

### 1.4. Study Aim

This study aimed to construct and factor-validate a shorter form of the WEIMS (Short Work Extrinsic and Intrinsic Motivation Scale; SWEIMS). First, the initial model of the SWEIMS will be constructed in employees in the construction industry [19,45]. Second, cross-validation of SWEIMS will be conducted in another sample of employees in the hospitality industry [16,17,18].

## 2. Methods

### 2.1. Standardisation Samples

The first sample was 155 construction workers in the UK (138 males and 17 females; Age range = 21–67, M = 40.28, SD = 11.05 years). The second sample consisted of 103 hospitality workers in the UK (47 males and 56 females; Age range 18–55, M= 28.2, SD = 8.6 years). These demographic data were similar to the general workforce sample of each industry [46,47,48]. Opportunity sampling methods were used to recruit the participants with a combination of online survey and paper-based survey. Eligible participants (a) were aged 18 years or older and (b) had been working for at least one year in the industry at the time of the study.

Those two samples were chosen because of the relevance to work motivation. Motivation of UK construction workers is challenged. They often work in irregular times, which hinders their work–life balance [19]. Moreover, the high physical risk work also compromises their work motivation [49]. Poor work motivation is one factor for high suicide rates in this industry [50,51]. Likewise, UK hospitality workers also struggle to stay motivated at work. Their work hours are often irregular, compromising employee work–life balance [17,18,52]. Hospitality work is characterised with emotional labour (e.g., dealing with pressure and stress while maintaining a professional display) that exhausts their work motivation [53]. Work motivation is relevant to these two industries. 

### 2.2. Ethics 

University research ethics committee approved this study. Informed consent was received from all participants before the study. No identifiable information was collected: anonymous data were collected.

### 2.3. Analysis 

Data were first screened for outliers and parametric test assumptions. Second, correlation analysis was used to determine two items from each of the WEIMS subscales. To avoid the risk of suboptimal content domain coverage [54], the contents of the selected items were reviewed by the researchers to ensure that those items for each type of work motivation captured the breadth of the original scale content [55]. Third, using data of the construction workers, the internal consistencies, and correlations between the same subscales in WEIMS and SWEIMS were assessed. These values were calculated in order to assess whether good internal consistencies were maintained in SWEIMS and correlations were similar between the two versions. These three steps regarded the construction of SWEIMS using the construction worker sample. The six-factor model of WEIMS was tested for SWEIMS through confirmatory factor analyses (CFAs), and cross-validation referring to the internal consistencies was performed in the hospitality worker sample. These steps related to the replication and factorial validation using the hospitality worker sample.

## 3. Results

### 3.1. Construction of SWEIMS Using Construction Worker Sample

In the WEIMS data collected from the construction workers, no outliers were detected using the boxplot (Appendix A) [56]. Data distribution was deemed normal assessed by the Q-Q plot. Correlation analysis identified two items from each work motivation type (12 items in total), and the contents were reviewed by all researchers ensuring that the original scale meaning was captured. This ensured that (a) the SWEIMS is correlated with the original WEIMS, and (b) the selected items represent their target subscale [57]. 

Table 1 shows the 12 selected items and their correlations with WEIMS subscales. Each individual item in SWEIMS demonstrated a correlation ranging between 0.82 and 0.92 with its target subscale, indicating very strong (*r* ≧ 0.80) correlations [58]. 

Internal consistencies for the WEIMS subscales ranged between 0.74 and 0.88, and that for SWEIMS ranged between 0.79 and 0.88, demonstrating acceptable (*α* ≧ 0.60) to high *(α* ≧ 0.80) reliability for WEIMS, and roughly high reliability for SWEIMS [55,59] (Table 2).

Correlations of the same subscales of the original WEIMS with those of the shorter form, SWEIMS were very strong [58]: *r* = 0.95 for Intrinsic Motivation, *r* = 0.96 for Integrated Regulation, *r* = 0.95 for Identified Regulation, *r* = 0.95 for Introjected Regulation, *r* = 0.96 for External Regulation, and *r* = 0.95 for Amotivation. 

### 3.2. Replication and Factorial Validation Using Hospitality Worker Sample

Model fit for the six-factor model, reported for the original WEIMS, was tested in the data collected from the hospitality worker sample (no outliers, normal distribution), using CFAs with RStudio version 13.30 (RStudio Team, 2020). The goodness of fit of the models was determined using the chi-squared to degrees of freedom ratio (χ^2^/df), the comparative fit index (CFI), the Tucker–Lewis Index (TLI), the root mean square error of approximation (RMSEA), and the standardised root mean residual (SRMR). All the assessment values indicated an adequate-to-good fit: χ^2^ = 68.25 (df = 39, p = 0.003), χ^2^/df = 1.75 (good fit [60]), CFI = 0.95, TLI = 0.92 (acceptable-to-good fit [61]), RMSEA = 0.085 (acceptable fit [62]), SRMR = 0.053 (good fit [61]). Figure 1 shows the factor structure (correlation matrices among the latent variables and factor loadings) of the six-factor model in SWEIMS.

Cross-validation was performed with the hospitality workers. The internal consistencies of WEIMS (0.66 to 0.82) and SWEIMS (0.65 to 0.78) were similar to each other, and all SWEIMS subscales had acceptable-to-high (≧0.60) reliability in this hospitality worker sample too (Table 3).

Compatible with the construction workers, the correlations between the original WEIMS and the shorter form SWEIMS in each subscale were also very strong in the hospitality workers: *r* = 0.94 for Intrinsic Motivation, *r* = 0.97 for Integrated Regulation, *r* = 0.95 for Identified Regulation, *r* = 0.89 for Introjected Regulation, *r* = 0.73 for External Regulation, and *r* = 0.97 for Amotivation.

## 4. Discussion

The present study aimed to develop a shorter form of the WEIMS [32]. Our analyses demonstrated that (a) the shorter form of WEIMS (SWEIMS) had strong correlations with the original WEIMS, (b) though slightly lower than the original WEIMS, the internal consistencies of SWEIMS were adequate to high, (c) the six-factor model was replicated in SWEIMS using CFAs, and (d) strong correlations between WEIMS and SWEIMS, and adequate to high internal consistencies of SWEIMS were found in both samples. These key findings suggest that the SWEIMS can be a reliable, valid and participant-friendly alternative for evaluating work motivation.

Measuring work motivation is important for several key reasons: SDT suggests that poor motivation is associated with poorer wellbeing, whereas highly motivated individuals experience greater wellbeing [63], contributing to a happier work environment. In addition to greater wellbeing, a highly motivated workforce is associated with greater retainment [64]. Highly motivated staff are more likely to engage in their jobs effectively and view them as positive and fulfilling [35]. Individuals with higher extrinsic and intrinsic motivation are also significantly more likely to invest greater time (i.e., working longer hours) and effort in their role [35]. Greater motivation and job engagement, which are significantly related to retention [16,17,65] which may therefore relate to lower expenditure and workplace costs. For example, the average cost of employee turnover in the UK is GBP 11000 per person [66]. Failure to address motivational issues can cost organisaitons. A brief and reliable work motivation scale is a valuable tool for many organisations. 

The SWEIMS can be used to measure motivation in a range of contexts. Organisational psychologists may wish to focus on motivation across a workforce, to highlight sections of an organisation or business which require improvement or additional resources/support, with the aim of improving efficiency. The SWEIMS includes six components of motivation (intrinsic motivation, integrated regulation, identified regulation, introjected regulation, external regulation and amotivation), which is detailed enough to provide an indication of how individuals within a workforce may be motivated. For instance, the questionnaire may highlight areas in which greater encouragement is needed; For individuals who are extrinsically motivated this may include workplace incentives or positive reinforcement such as guaranteed immediate financial benefits (healthcare contribution, life insurance meal allowances) and upskilling (professional training, private education; [67]). In individuals who are experiencing poor mental health at work, workplace counsellors may also address motivation, alongside workplace satisfaction, with the aim of understanding ways to improve workplace wellbeing and to discuss tailoring work to individuals who are motivated to perform well in specific tasks. Using this scale, workplace counsellors can understand an employee’s predominant motivation, and approach accordingly [68]. 

There is an increasing agenda to prioritise workplace wellbeing in the UK. As part of the UK Government Strategy on health, work and wellbeing, Public Health England have developed the Workplace Health Needs Assessment, to help employers to create healthier and productive work environments [69]. In comparison to the Work Extrinsic and Intrinsic Motivation Scale (WEIMS), the SWEIMS is much shorter, while still capturing different dimensions of motivation. Government initiatives could therefore consider including the SWEIMS in future workplace assessments considering minimal additional burden to staff. 

Finally, worldwide, workplaces have become more remote due to the coronavirus pandemic [70]. Using short questionnaires such as the SWEIMS to measure motivation is important to gain insight into the productivity and wellbeing of staff, particularly in the absence of relying on face-to-face behavioural cues.

Users of the SWEIMS need to be aware of implications for measuring work motivation using a questionnaire. Poor motivation at work may be underreported due to social desirability bias [27,71]. For example, in a masculine culture workplace, repressive coping may be present [19,45], where respondents unconsciously deny negative emotions, hindering honest responses. Social desirability bias can be reduced through minimising the presence of the researcher and using self-administered questionnaires [72]. Research using the SWEIMS should therefore consider recruiting participants anonymously and/or online, which may capture more accurate responses. Additionally recruiting participants away from everyday workplace settings may help to reduce the bystander effect in which individuals may be more likely to report socially desirable answers with a third-party presence [73]. Especially for workers in a culture where social presentation is highly valued such as Japan [74], such a consideration would help receive more accurate responses. 

Using the SWEIMS in addition to other data collection methods could obtain a broader understanding of workers’ motivation. Behavioural measures include participant observation of speed, performance and choice of tasks [71,75,76]. Cognitive measures of motivation such as memory accessibility and perceptions of goal-relevant objects may also be employed [76]. Qualitative interviews are also commonly used in work-motivation research to gain an in-depth understanding of specific groups within the workplace [77]. Considering the low workload of the SWEIMS, such combinations would be feasible. 

Despite the limitations of using questionnaires, there are practical benefits of using self-administered questionnaires including high practicality, and scalability to larger samples. Additionally, motivation may fluctuate over time [78] including crises such as pandemics [79]. Capturing these changes through repeated measurements is now more possible using the short 12-item version of the WEIMS which can be used frequently, while keeping participant burden to a minimal. As the importance of work motivation has been recognised alongside work mental health [80,81], the SWEIMS can offer an efficient and accurate way to assess work motivation.

### Limitations

This research has some limitations. Firstly, our samples only considered two industries: construction and hospitality. Further, the sample sizes were modest. Although evaluating consistency between two distinct samples is commonly practiced in this type of research [54], future research can include workers in other industries. Moreover, this study focused on the psychological aspect of work motivation, therefore did not consider socio-cultural aspects that may be relevant to work motivation. This was intended to prevent from creating other different variables between the samples, however future research can consider socio-cultural factors of work motivation [82]. Lastly, as the original version, interpretive thresholds are not established in WEIMS and SWEIMS. These markers can help maximise the utility of both WEIMS and SWEIMS.

## 5. Conclusions

Work motivation is important to many organisational outcomes including mental health. The WEIMS assesses six types of work motivation based on the SDT focusing on the autonomy of work motivation. The shorter 12-item version, SWEIMS developed in this study, needs less time and effort from busy workers, therefore is less susceptible to answer fatigue that can reduce the data quality. SWEIMS is a time-saving and user-friendly self-rating scale to assess work motivation.

## Figures and Tables

**Figure 1 ijerph-19-13864-f001:**
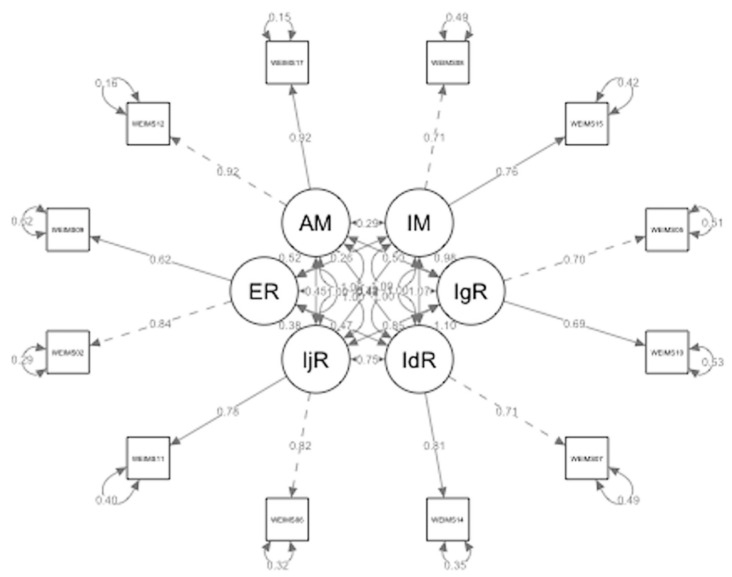
Factor structure of the six-factor model of the Short Work Extrinsic and Intrinsic Motivation Scale (SWEIMS). IM = Intrinsic Motivation; IgR = Integrated Regulation; IdR = Identified Regulation; IjR = Introjected Regulation; ER = External Regulation; AM = Amotivation.

**Table 1 ijerph-19-13864-t001:** Items for the Short Work Extrinsic and Intrinsic Motivation Scale (SWEIMS), including item-subscale correlations with subscale scores (155 construction workers).

No	SWEIMS Items	Subscale	*r*
8	For the satisfaction I experience from taking on interesting challenges	IM	0.92 **
15	For the satisfaction I experience when I am successful at doing difficult tasks.	IM	0.86 **
5	Because it has become a fundamental part of who I am.	IgR	0.89 **
10	Because it is part of the way in which I have chosen to live my life.	IgR	0.87 **
7	Because I chose this type of work to attain my career goals.	IdR	0.88 **
14	Because it is the type of work I have chosen to attain certain important objectives.	IdR	0.88 **
6	Because I want to succeed at this job, if not I would be very ashamed of myself.	IjR	0.82 **
11	Because I want to be very good at this work, otherwise I would be very disappointed.	IjR	0.90 **
2	For the income it provides me.	ER	0.92 **
9	Because it allows me to earn money.	ER	0.90 **
12	I don’t know why, we are provided with unrealistic working conditions.	AM	0.85 **
17	I don’t know, too much is expected of us.	AM	0.89 **

** *p* < 0.01. *r* = correlation coefficient. IM = Intrinsic Motivation; IgR = Integrated Regulation; IdR = Identified Regulation; IjR = Introjected Regulation; ER = External Regulation; AM = Amotivation.

**Table 2 ijerph-19-13864-t002:** Means, standard deviations and Cronbach’s alphas for the original Work Extrinsic and Intrinsic Motivation Scale (WEIMS) subscale scores and the Short Work Extrinsic and Intrinsic Motivation Scale (SWEIMS) subscale scores (155 construction workers).

	M		SD		*α*	
	WEIMS	SWEIMS	WEIMS	SWEIMS	WEIMS	SWEIMS
Intrinsic Motivation	4.57	4.66	1.56	1.64	0.84	0.84
Integrated Regulation	4.17	4.13	1.58	1.66	0.84	0.80
Identified Regulation	4.08	4.06	1.56	1.77	0.79	0.83
Introjected Regulation	4.20	4.25	1.55	1.65	0.81	0.79
External Regulation	4.66	4.77	1.70	1.75	0.88	0.88
Amotivation	2.71	2.90	1.28	1.54	0.74	0.81

**Table 3 ijerph-19-13864-t003:** Internal Consistencies for the Original Work Extrinsic and Intrinsic Motivation Scale (WEIMS) Subscale Scores and the Short Work Extrinsic and Intrinsic Motivation Scale (SWEIMS) Subscale Scores (103 Hospitality Workers).

	α	
	WEIMS	SWEIMS
Intrinsic Motivation	0.78	0.70
Integrated Regulation	0.82	0.65
Identified Regulation	0.78	0.72
Introjected Regulation	0.66	0.78
External Regulation	0.66	0.67
Amotivation	0.75	0.66

## Data Availability

The data that support the findings of this study are available on request from the corresponding author. The data are not publicly available due to privacy or ethical restrictions.

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
