# Peer review of "A Shorter Form of the Work Extrinsic and Intrinsic Motivation Scale: Construction and Factorial Validation"

_ijerph, 2022, doi:10.3390/ijerph192113864_

Round 1

Reviewer 1 Report

ID: ijerph-1983075

Title: A shorter form of the Work Extrinsic and Intrinsic Motivation Scale: Construction and factorial validation

Thank you for providing a chance to review this manuscript.

Comment: Major Revision.

Detailed information:

Abstract

Line 18-22 Page 1: 1) What does the shortened form of WEIMS contain? 2) Explain how you constructed and validated. 3)The shorter form still examines all of six types of work motivation being mentioned in WEIMS or not?  And which field be shorten in the shorter form, concerning the whole scale’s integrity?

Line 22-25, Page 1: 1) Please describe the characteristics of the samples, including number, occupation, etc. 2) What is your statistical method? 3) Please list statistical values for internal consistency and correlation analysis. 4) What about the validity analysis?

It is recommended that the author rewrite the abstract section. There is a lot of important information missing. Please refer to published articles in this journal.

Introduction

Work motivation as key for workplace mental health

Line 32, Page 1: Actually, much more than I had ever expected references in this part, it will be better if using refined language to summary unimportant and uncorrelated references.

Line 52-57, Page 2: “Those with lower work motivation tended to feel more shame about their experience of work-related mental health problems compared to those with higher work motivation, suggesting those with lower work motivation may identify with their work less, and thus identify with and take personal responsibility for their mental health issues more (e.g., ‘I don’t invest much to my work, therefore I should have better mental health’)”. The expression of this sentence is difficult to understand, please reorganize the language.

Measuring work motivation

Line 104-105, Page 3: “Although some evidence has questioned the SDT continuum model underlying the WEIMS [41], its reliability and validity across wide-ranging organisational contexts positions it as a robust instrument to assess workplace motivation”. What kind of doubt? More evidence should be given to support your opinion.

Line 113-121, Page 3: 1) How much time does it take to complete a WEIMS, how much time does SWEIMS take? The time to complete the scale was not studied in your research! 2) What is innovation of your study?

Methods

Standardisation samples

Line 132-135, Page 3: 1) Why did you choose these two occupations? In addition, it is not enough to validate the scale in these two occupations, nor is the number of samples. 2) The male-female ratio in construction workers is lopsided, how to control the gender bias? 3) How to ensure that online survey and paper survey have the same effectiveness? I'm suspicious of your investigative methods. 4) Please specify the inclusion and exclusion criteria for the samples.

Analysis

Line 142-143, Page 3: What is the purpose of correlation analysis in WEIMS subscales and what is the next step?

Line 146, Page 4: Were only CFAs used in the validity analysis? What about content validity?

Line 147, Page 4: Does cross-validation only verify internal consistency? Is there cross-validation of validity?

Line 148, Page 4: How to define the significance of statistical analysis?

Overall: The shorten process and validation of WEIMS should be described in detail in “Methods”! The author puts a lot of information in the “Results”. Please review the article and rewrite the method!

Results

Construction of SWEIMS using construction worker sample

This part should be in the “Methods”! The construction method of SWEIMS is far from enough. What are the dimensions, items and scoring range of the scale? Please elaborate on your adaptation!

Line 151-159, Page 4: 1) Was the adaptation of WEIMS approved by the original author? 2) Why use the data from construction worker? 3) “The initial version of SWEIMS was constructed using correlation analysis.” The construction of SWEIMS is confusing. Please elaborate, this is very important content!

Line 152, Page 4: “No outliers were detected using the boxplot.” Show the boxplot diagram in the article.

Line 160-162, Page 4: How relevant are the other items to WEIMS? In addition, the SWEIMS items are all originated from WEIMS, so of course the relevance is high. I don't think the analysis you do can explain anything.

Replication and factorial validation using hospitality worker sample

Line 182, Page 7: Are there any outliers?

Overall: 1) All tables should be presented as three-line tables; 2) The description of the results needs to list the p-values.

Discussion

Line 220-230: Page 7: This is a repetition of the introduction and has little to do with your findings.

Page 8: As a scale, it’s a better choice to add a discussion about popularization.

Overall: 1) Regarding methods, it's not clear what has been utilized from the existing work and what is new (proposed by the authors). It should be clearly discussed. 2) All insignificant discussions and quotes must be deleted from the text. You spend a lot of readers' time with irrelevant or obvious details. The current form of the introduction is inappropriate; it's long and irrelevant. Discussion should focus on your results and methods, not on the background.

Until here, with all due respect, I did not even finish all the reading. First, reading more articles from the TOP journals, to learn the formats, expressions, and of great importance—logic, might help a lot before revising. Second, the process of scale adaptation and verification should be clearly written in the method, and the results should clearly list the meaningful results. Last, rephrase your sentences to make your expressions clear, especially the introduction and discussion sections. Some of your sentences and paragraphs are irrelevant, and hard to read and understand. As I have repeatedly emphasized, logic plays a key role in scientific writing. Making your valuable data an intriguing story to tell and a coherent article for people to read, is of great importance. Furthermore, finding a native English speaker to improve the writing can considerably improve the quality.

Thank you and my best,

Your reviewer

Author Response

Response Letter

Manuscript ID: ijerph-1983075

"A shorter form of the Work Extrinsic and Intrinsic Motivation Scale: Construction and factorial validation”

Dear Reviewers,

Thank you for your helpful feedback. We have systematically revised our manuscript addressing the points you have raised. Please see our responses below. We hope this revised paper is now acceptable for publication. We extend our sincere gratitude to you for your feedback that has significantly helped to strengthen the paper.

Reviewer 1

Reviewer 1’s comment 1

Abstract

Line 18-22 Page 1: 1) What does the shortened form of WEIMS contain? 2) Explain how you constructed and validated. 3)The shorter form still examines all of six types of work motivation being mentioned in WEIMS or not?  And which field be shorten in the shorter form, concerning the whole scale’s integrity?

Line 22-25, Page 1: 1) Please describe the characteristics of the samples, including number, occupation, etc. 2) What is your statistical method? 3) Please list statistical values for internal consistency and correlation analysis. 4) What about the validity analysis?

It is recommended that the author rewrite the abstract section. There is a lot of important information missing. Please refer to published articles in this journal.

Authors’ response 1-1

Thank you for your helpful feedback. Now the abstract has been re-written addressing your points.

Reviewer 1’s comment 2

Introduction

Work motivation as key for workplace mental health

Line 32, Page 1: Actually, much more than I had ever expected references in this part, it will be better if using refined language to summary unimportant and uncorrelated references.

Line 52-57, Page 2: “Those with lower work motivation tended to feel more shame about their experience of work-related mental health problems compared to those with higher work motivation, suggesting those with lower work motivation may identify with their work less, and thus identify with and take personal responsibility for their mental health issues more (e.g., ‘I don’t invest much to my work, therefore I should have better mental health’)”. The expression of this sentence is difficult to understand, please reorganize the language.

Authors’ response 1-2

In line with your comment, this part of the introduction is re-worked, and significantly reduced in its length.

Reviewer 1’s comment 3

Measuring work motivation

Line 104-105, Page 3: “Although some evidence has questioned the SDT continuum model underlying the WEIMS [41], its reliability and validity across wide-ranging organisational contexts positions it as a robust instrument to assess workplace motivation”. What kind of doubt? More evidence should be given to support your opinion.

Line 113-121, Page 3: 1) How much time does it take to complete a WEIMS, how much time does SWEIMS take? The time to complete the scale was not studied in your research! 2) What is innovation of your study?

Authors’ response 1-3

In line with your comment, the details of the doubt on the SDT continuum. We believe the second point relates to the fact that we did not clarify how the SWEIMS shortened the original WEIMS. This is a better explanation than the length of time required to complete, as it varies by person. This is now added. Thank you for your insights.

Reviewer 1’s comment 4

Methods

Standardisation samples

Line 132-135, Page 3: 1) Why did you choose these two occupations? In addition, it is not enough to validate the scale in these two occupations, nor is the number of samples. 2) The male-female ratio in construction workers is lopsided, how to control the gender bias? 3) How to ensure that online survey and paper survey have the same effectiveness? I'm suspicious of your investigative methods. 4) Please specify the inclusion and exclusion criteria for the samples.

Authors’ response 1-4

In line with your comment, a justification for choosing those two samples is added. Eligibility criteria are now added. The gender balance for the construction workers corresponds to the general sample of UK construction workers. This is now explained. However, as you indicated, more samples are needed to confirm our findings. This is now added to the limitation. Moreover, the difference between online survey and paper survey is now added to the limitation, and future research.

Reviewer 1’s comment 5

Analysis

Line 142-143, Page 3: What is the purpose of correlation analysis in WEIMS subscales and what is the next step?

Line 146, Page 4: Were only CFAs used in the validity analysis? What about content validity?

Line 147, Page 4: Does cross-validation only verify internal consistency? Is there cross-validation of validity?

Line 148, Page 4: How to define the significance of statistical analysis?

Overall: The shorten process and validation of WEIMS should be described in detail in “Methods”! The author puts a lot of information in the “Results”. Please review the article and rewrite the method!

Authors’ response 1-5

In line with your comment, now the purpose of correlation analysis and CFAs are clarified. Content validity was done by ensuring that the selected items for each type of work motivation captured the breadth of the original scale content. This is to mitigate the risk of suboptimal content domain coverage. The relevant parts are moved from the results section to the methods.

We have revisited published papers about construction and factorial validation to make sure our methods are robust:

https://pubmed.ncbi.nlm.nih.gov/21584907/

https://doi.org/10.1080/03069885.2021.1903387

https://doi.org/10.1080/13674676.2022.2114441 (in press)

Reviewer 1’s comment 6

Results

Construction of SWEIMS using construction worker sample

This part should be in the “Methods”! The construction method of SWEIMS is far from enough. What are the dimensions, items and scoring range of the scale? Please elaborate on your adaptation!

Line 151-159, Page 4: 1) Was the adaptation of WEIMS approved by the original author? 2) Why use the data from construction worker? 3) “The initial version of SWEIMS was constructed using correlation analysis.” The construction of SWEIMS is confusing. Please elaborate, this is very important content!

Line 152, Page 4: “No outliers were detected using the boxplot.” Show the boxplot diagram in the article.

Line 160-162, Page 4: How relevant are the other items to WEIMS? In addition, the SWEIMS items are all originated from WEIMS, so of course the relevance is high. I don't think the analysis you do can explain anything.

Replication and factorial validation using hospitality worker sample

Line 182, Page 7: Are there any outliers?

Overall: 1) All tables should be presented as three-line tables; 2) The description of the results needs to list the p-values.

Authors’ response 1-6

In line with your comment, only necessary information that relates to the results is now left in the results section. As with other scales and the original WEIMS validation paper, approval was not needed to validate the short version. A justification for using these two samples was added.

The boxplots are submitted as appendices, as the relevance of these outputs to the study and the space required are not proportionate.

No outliers were detected in the hospitality sample. This is now added.  Tables are adjusted following the journal guidelines.

Reviewer 1’s comment 7

Discussion

Line 220-230: Page 7: This is a repetition of the introduction and has little to do with your findings.

Page 8: As a scale, it’s a better choice to add a discussion about popularization.

Authors’ response 1-7

Following the journal guidelines, we will keep a summary of the findings at the beginning of the discussion. In line with your comment, popularization is embedded.

Reviewer 2 Report

This study develops a shortened version of motiation scale (i.e. WEIMS) and examines the validity of the scales using two different samples. I think the research makes sense to me and the scale can bring a more efficient way to capture motivation. However, the reserach motivation should be futher elaborated, in terms of why this research should be done and what value we can find beyond efficiency. Furthermore, the authors may want to explain why this study attemtps to examines the scales with two particular contexts. The sample selection selection procedure should be clearly stated.

Hope these comments help to revise the paper.

Author Response

Response Letter

Manuscript ID: ijerph-1983075

"A shorter form of the Work Extrinsic and Intrinsic Motivation Scale: Construction and factorial validation”

Dear Reviewers,

Thank you for your helpful feedback. We have systematically revised our manuscript addressing the points you have raised. Please see our responses below. We hope this revised paper is now acceptable for publication. We extend our sincere gratitude to you for your feedback that has significantly helped to strengthen the paper.

Reviewer 2

Reviewer 2’s comment 1

This study develops a shortened version of motiation scale (i.e. WEIMS) and examines the validity of the scales using two different samples. I think the research makes sense to me and the scale can bring a more efficient way to capture motivation. However, the reserach motivation should be futher elaborated, in terms of why this research should be done and what value we can find beyond efficiency. Furthermore, the authors may want to explain why this study attemtps to examines the scales with two particular contexts. The sample selection selection procedure should be clearly stated.

Authors’ response 2-1

Thank you for your helpful feedback. In line with your comments, now an explanation for those 2 samples and the process is now added.

Reviewer 3 Report

This work aimed to construct and factor-validate a shorter form of the WEIMS (Short Work Extrinsic and Intrinsic Motivation Scale; SWEIMS), due to the original WEIMS was too long for busy people at work, by evaluating the correlation between work motivation and workplace mental health, SWEIMS proved its reliability, validity, and user-friendliness. The topic is attractive. Some comments for the authors to improve the quality of the manuscript are below.

1. In line 125-128, why were the construction workers chosen for initial model and the hospitality industry workers chosen for the cross-validation of the SWEIMS? Could the author give some explanations? Why did the authors not recruit another group of construction workers to cross-validate the SWEIMS?

2. The sample sizes of the two groups of workers were too small (i.e. 155 and 103). I think it is not enough to generate robust results.

3. The text expression for the section of Methods is a bit concise: how to define the outliers of samples? What were the parametric test assumptions and cross-validation? How was the scale format? Lacking a more detailed explanation. Also, the method part was not clearly written to provide the details of the data analysis. For example, the convergent and discriminant validity; and test-retest reliability were not assessed. The authors may refer to the work of the CoWoRP scale development in the literature.

4. In Table 1, it shows us the 12 items chosen of the SWMIES, compared to the original WMIES, what were the reasons that you choose them for your scale?

5. In the section of Results, the placement of the graphs lacks standard, needed to adjust the sizes and positions. Besides, Figure 1 is not legible enough to see details.

6. The limitation of SWMIES was better to have a separate subheading.

7. The formatting issue in line 212.

8. How to apply the SWMIES to industries is not clear. What is its cut-off value to determine if workers are highly motivated?

Author Response

Response Letter

Manuscript ID: ijerph-1983075

"A shorter form of the Work Extrinsic and Intrinsic Motivation Scale: Construction and factorial validation”

Dear Reviewers,

Thank you for your helpful feedback. We have systematically revised our manuscript addressing the points you have raised. Please see our responses below. We hope this revised paper is now acceptable for publication. We extend our sincere gratitude to you for your feedback that has significantly helped to strengthen the paper.

Reviewer 3

Reviewer 3’s comment 1

This work aimed to construct and factor-validate a shorter form of the WEIMS (Short Work Extrinsic and Intrinsic Motivation Scale; SWEIMS), due to the original WEIMS was too long for busy people at work, by evaluating the correlation between work motivation and workplace mental health, SWEIMS proved its reliability, validity, and user-friendliness. The topic is attractive. Some comments for the authors to improve the quality of the manuscript are below.

  1. In line 125-128, why were the construction workers chosen for initial model and the hospitality industry workers chosen for the cross-validation of the SWEIMS? Could the author give some explanations? Why did the authors not recruit another group of construction workers to cross-validate the SWEIMS?

Authors’ response 3-1

Thank you for your helpful feedback. In line with your comments, now an explanation for the sample selection is added. Cross-validation benefits from comparing different samples, thus we tested in the hospitality worker sample.

Reviewer 3’s comment 2

  1. The sample sizes of the two groups of workers were too small (i.e. 155 and 103). I think it is not enough to generate robust results.

Authors’ response 3-2

In line with your comments, now this is added to the limitation.

Reviewer 3’s comment 3

  1. The text expression for the section of Methods is a bit concise: how to define the outliers of samples? What were the parametric test assumptions and cross-validation? How was the scale format? Lacking a more detailed explanation. Also, the method part was not clearly written to provide the details of the data analysis. For example, the convergent and discriminant validity; and test-retest reliability were not assessed. The authors may refer to the work of the CoWoRP scale development in the literature.

Authors’ response 3-3

In line with your comments, methods for detecting outliers and for testing parametric assumptions are added. Thank you for your suggestion regarding the scale development, however our study focused on shortening the existing scale. To make this clear, the wording throughout the paper is adjusted to assure the intent of this paper is expressed.

Reviewer 3’s comment 4

  1. In Table 1, it shows us the 12 items chosen of the SWMIES, compared to the original WMIES, what were the reasons that you choose them for your scale?

Authors’ response 3-4

In line with your comments, reasons for choosing those 12 are now added.

Reviewer 3’s comment 5

  1. In the section of Results, the placement of the graphs lacks standard, needed to adjust the sizes and positions. Besides, Figure 1 is not legible enough to see details.

Authors’ response 3-5

In line with your comments, we have revisited the journal guidelines and adjusted accordingly. An improved figure is now inserted.

Reviewer 3’s comment 6

  1. The limitation of SWMIES was better to have a separate subheading.

Authors’ response 3-6

Now the subheading is added.

Reviewer 3’s comment 7

  1. The formatting issue in line 212.

Authors’ response 3-7

Now the formatting is corrected.

Reviewer 3’s comment 8

  1. How to apply the SWMIES to industries is not clear. What is its cut-off value to determine if workers are highly motivated?

Authors’ response 3-8

Thank you for your helpful question. This is not determined in the original WEIMS either, suggesting that more comprehensive data collection is needed for both versions. Otherwise, as you indicated, the utility of these scales won’t be maximised. This is added in the limitation.

Round 2

Reviewer 1 Report

ID: ijerph-1983075

Title: A shorter form of the Work Extrinsic and Intrinsic Motivation Scale: Construction and factorial validation

Most of the questions have been revised and supplemented to make the article more rigorous and scientific. I have some further suggestions for improving the manuscript.

Detailed information:

Methods

Standardisation samples

I still think it is inappropriate to just select construction workers, which is the key flaw in this study! These reasons cannot convince me.

Results

The form of the form is still problematic, please review it.

Thank you and my best,

Your reviewer

Author Response

Response Letter 2

Manuscript ID: ijerph-1983075

"A shorter form of the Work Extrinsic and Intrinsic Motivation Scale: Construction and factorial validation”

Dear Reviewers,

Thank you again, for your helpful feedback. We have systematically revised our manuscript addressing the points you have raised. Please see our responses below. We hope this revised paper is now acceptable for publication. We extend our sincere gratitude to you for your feedback that has significantly helped to strengthen the paper.

Reviewer 1

Reviewer 1’s comment

Detailed information:

Methods

Standardisation samples

I still think it is inappropriate to just select construction workers, which is the key flaw in this study! These reasons cannot convince me.

Results

The form of the form is still problematic, please review it.

Authors’ response 1

Thank you for your helpful feedback. Now clarity is added to the standardisation samples that these include construction and hospitality samples. Also this is added to the limitation section. 

Thank you, now the tables are re-formatted. 

Reviewer 3 Report

This modification has improved most of the problems, but there are still some minor problems that need to be adjusted:

1.In Figure 1, the text inside the picture is still unreadable, and the numbers and lines in the middle overlap together too much, so I suggest enlarging the middle part and shortening the lines around it to improve the clarity of the picture.

2.In Appendix 1, it is recommended to align and resize all the charts, which will look more standardized and uniform.

Author Response

Response Letter 2

Manuscript ID: ijerph-1983075

"A shorter form of the Work Extrinsic and Intrinsic Motivation Scale: Construction and factorial validation”

Dear Reviewers,

Thank you again, for your helpful feedback. We have systematically revised our manuscript addressing the points you have raised. Please see our responses below. We hope this revised paper is now acceptable for publication. We extend our sincere gratitude to you for your feedback that has significantly helped to strengthen the paper.

Reviewer 3

Reviewer 3’s comment

1.In Figure 1, the text inside the picture is still unreadable, and the numbers and lines in the middle overlap together too much, so I suggest enlarging the middle part and shortening the lines around it to improve the clarity of the picture.

2.In Appendix 1, it is recommended to align and resize all the charts, which will look more standardized and uniform.

Authors’ response 1

Thank you for your helpful feedback. Now the figure is revised as suggested. The main function of this figure is to show that the six-factor model was replicated. This figure now demonstrates the model. 

Appendix has been revised, thank you.